# Data-dependent visualization of biological networks in the web-browser with NDExEdit

**Florian Auer** *, **Simone Mayer**, **Frank Kramer**

Department of IT-Infrastructure for Translational Medical Research, Faculty of Applied Computer Science, University of Augsburg, Augsburg, Germany

* florian.auer@informatik.uni-augsburg.de

## Abstract

Networks are a common methodology used to capture increasingly complex associations between biological entities. They serve as a resource of biological knowledge for bioinformatics analyses, and also comprise the subsequent results. However, the interpretation of biological networks is challenging and requires suitable visualizations dependent on the contained information. The most prominent software in the field for the visualization of biological networks is Cytoscape, a desktop modeling environment also including many features for analysis.

A further challenge when working with networks is their distribution. Within a typical collaborative workflow, even slight changes of the network data force one to repeat the visualization step as well. Also, just minor adjustments to the visual representation not only need the networks to be transferred back and forth. Collaboration on the same resources requires specific infrastructure to avoid redundancies, or worse, the corruption of the data. A well-established solution is provided by the NDEx platform where users can upload a network, share it with selected colleagues or make it publicly available.

NDExEdit is a web-based application where simple changes can be made to biological networks within the browser, and which does not require installation. With our tool, plain networks can be enhanced easily for further usage in presentations and publications. Since the network data is only stored locally within the web browser, users can edit their private networks without concerns of unintentional publication. The web tool is designed to conform to the Cytoscape Exchange (CX) format as a data model, which is used for the data transmission by both tools, Cytoscape and NDEx. Therefore the modified network can be directly exported to the NDEx platform or saved as a compatible CX file, additionally to standard image formats like PNG and JPEG.

## Author summary

Relations in biological research are often visualized as networks. For instance, if two proteins interact with each other during a certain process, the corresponding network would show two nodes connected by one edge. But the fact that the interaction between the two exists, may not be enough. With established software solutions like Cytoscape we can add

**Data Availability Statement:** A live demo is hosted on GitHub Pages at https://frankkramer-lab.github.io/NDExEdit and the corresponding source code for deploying own instances is provided at https://github.com/frankkramer-lab/NDExEdit.

**Funding:** This work was supported by the Multipath project (https://www.sys-med.de/en/junior-research-groups/multipath/) funded by the German Ministry of Education and Research (Bundesministerium für Bildung und Forschung, BMBF) through grants to FK (grant FKZ01ZX1508). The funders had no role in study design, data collection and analysis, decision to publish, or preparation of the manuscript.

**Competing interests:** The authors have declared that no competing interests exist.

all the information we have about our nodes and their interaction to our data foundation. Furthermore, we can change the visual appearance of our nodes and their interaction based on this information.

For example, if our network contains 20 nodes, that all interact with each other, but the strength of these interactions each range between 0 and 1, we can illustrate that by making the edges wider for strong interactions and slimmer for weak interactions. Thus, our visualization is enriched with valuable information. As of now these data-dependent modifications can only be made with a desktop client.

We introduce NDExEdit, a web-based solution for visualization changes to networks that conform to the CX data format. It allows us to import networks directly from the NDEx platform and apply changes to the visualization—including all types of mappings, one of which was briefly described above.

This is a *PLOS Computational Biology* Software paper.

## Introduction

Networks are well-established in a wide range of fields in biology [1–3], and are often used, either as a source or result, in biological research. Information associated with the individual nodes or edges can go far beyond name and type, thus increasing its complexity. Within common bioinformatics workflows data integration, network analysis, and visualization accompany each other [4, 5], and comprise fundamental challenges of combining various tools.

The information-rich data contained in biological networks provide the opportunity for comprehensive visualization but requires powerful tools to achieve. Cytoscape [6] is the most prominent desktop software for biological network analysis and visualization. It employs a data-dependent visualization strategy by applying so-called "attribute-to-visual-mappings", where a node's or edge's attribute translates to its visual representation. Besides its support for large networks and its rich set of features, Cytoscape comes with overhead for quick results and a steep learning curve.

A major challenge when working with networks is their distribution. Collaboration on the same resources requires specific infrastructure to avoid redundancies, or worse, the corruption of the data. A well-established solution is provided by the NDEx platform [7, 8] where users can upload a network, share it with selected colleagues or make it publicly available. NDEx also holds the feature to provide your private networks solely to the reviewers of a submitted paper, to protect the data until publication.

NDEx is tightly connected to Cytoscape, which reveals itself in the mutual integration of both platforms. For the transmission of the networks the Cytoscape Exchange (CX) data structure [9] was developed, which not only includes the structural information of the networks but also instructions for its visual representation.

There is a recent trend in software development towards web-based solutions. Desktop applications require individual installations, which is not possible in all cases for various reasons and also brings further expense for maintenance. Furthermore, accessibility across different devices grows in importance, while web-based applications provide secure access to centralized data. In the following, we illustrate how our lightweight web application NDExEdit implements current web technologies and thereby facilitates the data-dependent visualization of biological networks.

## Design and implementation

### Network data model

CX is a JSON (JavaScript Object Notation) based data structure designed for the transmission of biological networks between web applications and servers. The different types of information within a network are organized into single aspects of the network. These modular components separate the basic network structure from additional information and thus enable to only load the parts of the network that are of interest for an application. Since CX is designed as a transmission format, this reduces the amount of data needed to be transferred, but still combines all data as one coherent network.

The aspects have a defined scheme for the elements they can contain that must be followed. This includes definitions for core aspects, concerning the network topology and attributes, and aspects contributed by Cytoscape handling the visual representation. They link to each other by referencing the internal ID used in the aspects, for example, refer edges the IDs of the nodes aspects they are connecting. Furthermore, it is possible to include own custom aspects without a strict definition, that will be stored at the NDEx platform, but not processed or validated.

### Implementation details

The client-side visualization of networks is realized using Cytoscape.js [10]. It is a JavaScript library for browser and server-based graph rendering, including layout algorithms for positioning nodes. One of its key features is the separation of data and its representation: stylesheets are used to data-dependently select network elements and assign visual properties to them.

Cytoscape.js does not natively support the handling of networks in CX format but is used in the front-end of the NDEx platform to visualize the CX networks. Their mapping script was incorporated into NDExEdit to assure a consistent visual representation in all software tools, including Cytoscape. Therefore, modifications of the script were necessary to enable highlighting and export of the networks.

The functionality of NDExEdit rests upon the Angular [11] platform, an open-source framework for building single-page web applications. It follows the Model-View-Controller (MVC) design pattern which reduces the code required for implementing the web application. Angular is based on TypeScript [12] as the programming language, which brings advantages for development in form of static typing and support of class-based object-oriented programming (OOP).

The layout of the web application is realized using the Bootstrap [13] framework. It is an open-source CSS framework for front-end development, containing design templates for interface components.

## Results

NDExEdit simplifies the visual adjustment of networks and illustrates the great potential of web-based solutions for biological research: Users with any operating system can work with NDExEdit without a requirement for installation or account. Since the installation of desktop clients is often restricted due to security concerns, web-based applications can close this gap and provide access through mobile devices. It runs only in the web browser, without any supporting backend infrastructure, which ensures data privacy while still providing flexibility in the visualization workflow. Those concerns can even be reduced further by setting up private installations and securing their accessibility.

The web application provides a lightweight interface to explore the contents of networks and facilitates the quick defining of custom visualizations dependent on the data. Networks can be layouted using a variety of built-in algorithms, and refined manually. With compliance to the Cytoscape Exchange format, the network data and its visualization is contained within the same resource, which representation also remains consistent between all tools. NDExEdit narrows the gap between desktop software to create and edit a network, and web-based platforms to decorate and distribute them.

## Web-application

A typical workflow within NDExEdit starts with the import of networks, for which several options are provided: The user can browse and query the publicly available, or by supplying personal credentials also the own private networks on the NDEx platform, and load selected ones directly into the app. Alternatively, networks can be loaded from a provided NDEx UUID or URL, or a local CX file. All successfully imported networks become accessible in the overview list and are ready for modification. The home button of any subordinate page leads back to this page to be able to switch between networks.

By default, the breast cancer protein-protein interaction network by Minkyu Kim [14] is provided for demonstration purposes. The network contains the interactome of all high-confidence PPIs detected across the three breast cell lines MCF7, MDA-MB-231, and MCF10A. Besides the valuable information contained in this network, it is also a great example of how the visual representation (Fig 1) supports the comprehension of the underlying data. Therefore, it will be used in the following to demonstrate the capabilities of NDExEdit to define and edit the attribute mappings dependent on the network data.

When accessing a network in NDExEdit, general information about it will be shown next to its visualization. This view can be customized by toggling the sides or moving the separating border in any direction. The general information panel provides an overview of all node, edge, and network attributes of the network. While the network attributes can be edited directly, the remaining attributes can be explored for their distribution and the coverage of the nodes and edges by this attribute. Additionally, the network can be inspected by creating rules on the values of the node and edge attributes to be highlighted in the graph.

The visualization of the network is interactive, which means that it can be zoomed and shifted, and also the nodes and edges can be selected and moved. Detailed information about the selected elements appears on top in the information panel to be able to compare its content. With the available buttons, the graph can be fit to the viewport and for better overview and performance improvements, the labels in the network can be hidden.

## Attribute mappings

A key feature within the data-dependent visualization in Cytoscape is the so-called "attribute-to-visual-mappings" where the values of an attribute are processed by a specific function to generate a new value for the visual representation. Thereby one attribute (or property in the CX context) can be mapped to several visual properties. Cytoscape and the CX-file format distinguish between three kinds of mapping types that can be applied to nodes as well as edges: discrete, continuous, and pass-through.

The values of a property can vary in its data type, which limits the types of mappings that can be applied. For example, for string values, it is not possible to apply a continuous mapping, since by its nature only discrete manifestations are given without any order.

On the other hand, the visual properties vary by type of the value to which they are mapped:

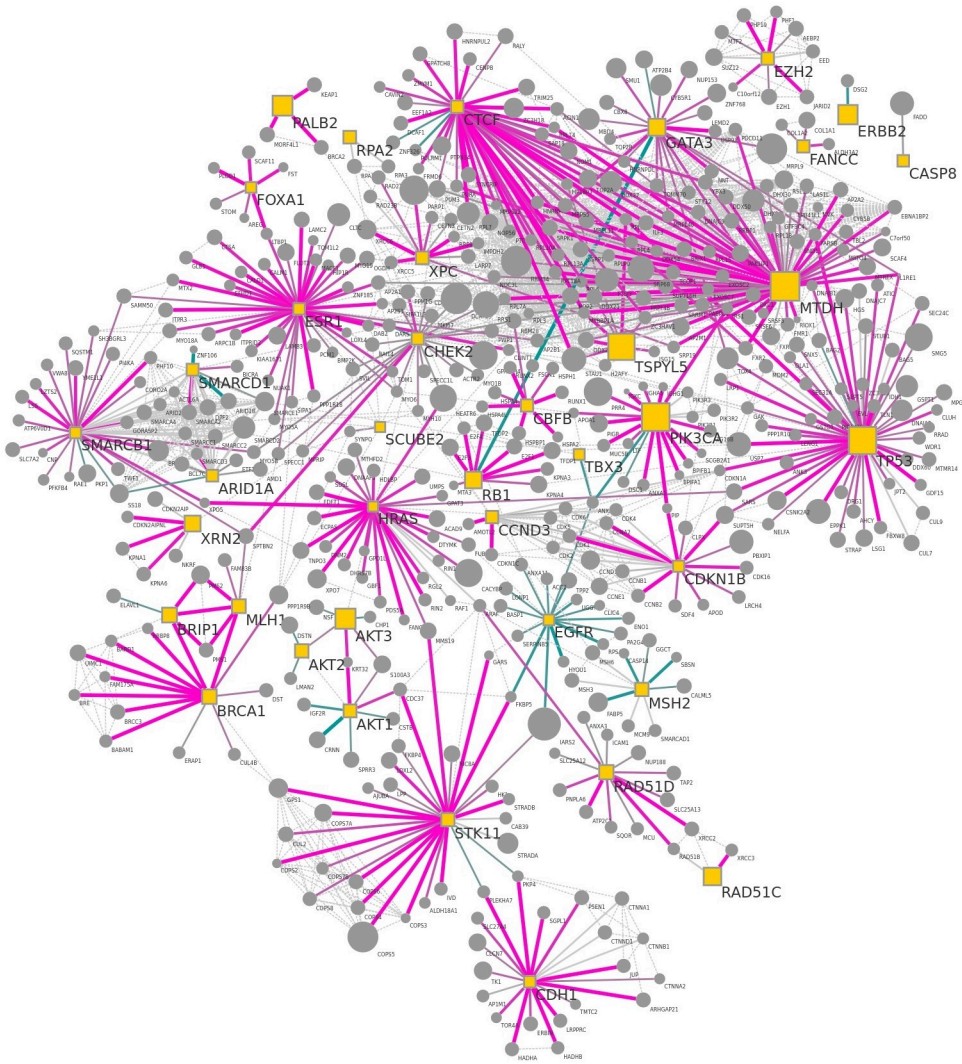

**Fig 1. Breast cancer protein-protein interaction network used as example network on NDExEdit.** It shows the interactome of the union of all high-confidence PPIs detected across breast cell lines MCF7, MDA-MB-231, and MCF10A. This network is available on NDEx by the UUID: e89ad762-ab4b-11ea-aaef-0ac135e8bacf.

- numerical values for example "NODE_SIZE" or "EDGE_WIDTH"

- colors (in hexadecimal format) for example "NODE_FILL_COLOR"

- string values as in "ELLIPSE" for a "NODE_SHAPE"

- font declarations, including font-family, -style, and -size, are used for example for "NODE_LABEL_FONT_FACE"

NDExEdit limits the choices to select for visual mapping properties to only the applicable types to assure, that only valid mappings can be created. Custom selection tools for colors and fonts are included as well to facilitate the creation of new mappings. The highlighting of attributes and modification of the mappings does not take effect immediately to prevent disruptive errors in the data model and the visualization. Instead, the modification of other attributes is

locked and visually indicated by warning signs on the superior elements and a surrounding frame.

The mappings themselves are stored within the network in the "cyVisualProperties" aspect. This ensures a consistent visual representation of the network on all three platforms, namely NDExEdit, Cytoscape, and NDEx. Furthermore, the modification of the mappings can be continued on either NDEx or Cytoscape.

**Discrete mapping.** Discrete mappings are the most straightforward type of mappings: to one discrete value of a property, a corresponding mapping value is explicitly assigned. This way, all manifestations of the property can be set individually, but also left blank if no or a default value should be used. Fig 2 shows the discrete mappings of the provided sample network for the properties "Bait" and "BaitBoolean". It shows that each property has only one possible value with already several mappings to visual properties of different data types.

The mapping for the "Bait" property is shown in editing mode with an additional visual property already added using the green plus symbol next to it. The missing mapping value can easily be added using the gray plus symbol or removed with the red "X" button. Also, the visual properties can be removed or restored to the initial value before editing via the provided buttons.

The applied changes can be tested by temporarily showing their effects in the graph by using the magic wand button. All made adjustments can be omitted through the red "X" at the bottom, which leads back to the network overview. Only by actively accepting the changes the new mapping is applied and saved for export.

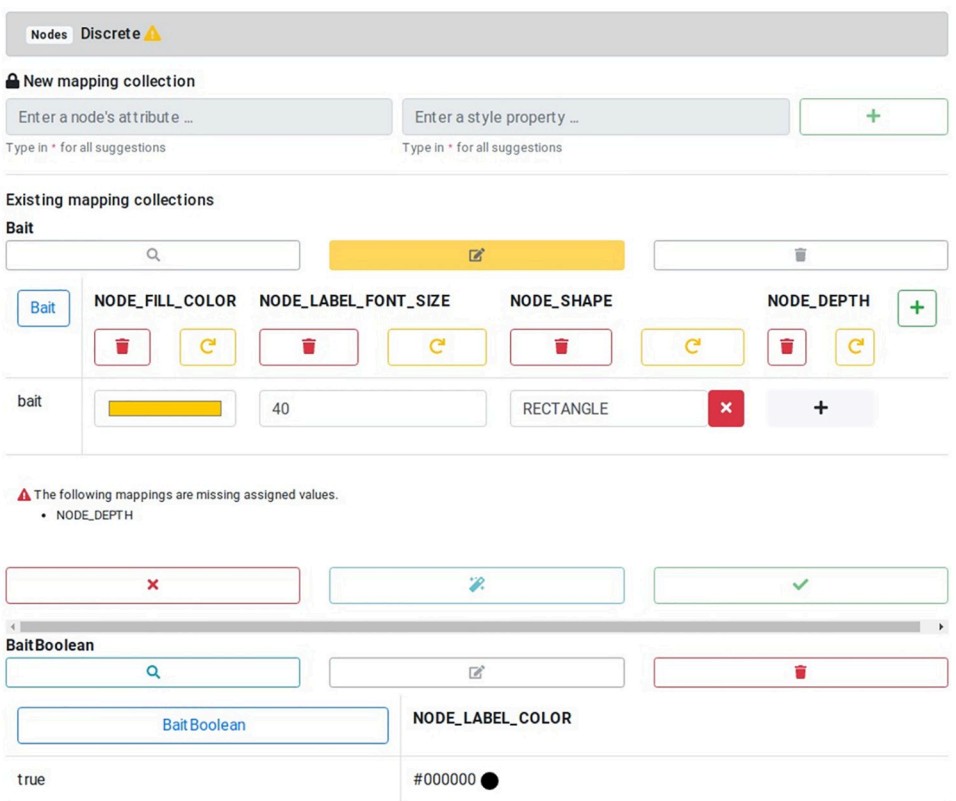

**Fig 2. Discrete mapping for node properties.** New discrete mappings can be created, existing mappings are shown for the "Bait" property of several visual properties. This includes mappings to colors, numerical and concrete string values.

**Continuous mapping.**   Defining a discrete mapping for continuous values would be tedious since for every value occurring in the attribute a corresponding value for the visual property would be needed. Continuous mappings relieve one from this burden by defining a function on which basis the values for the visual properties are generated. This function is simply characterized by thresholds for the attribute values with corresponding values for the visual property. All values between two thresholds are then mapped linearly in-between.

Continuous mappings can be defined in NDExEdit similarly as discrete mappings, only that the thresholds have to be defined first. Fig 3 shows the continuous mapping of the "diff_-score" attribute to two visual properties of the edges. Although several thresholds are defined,

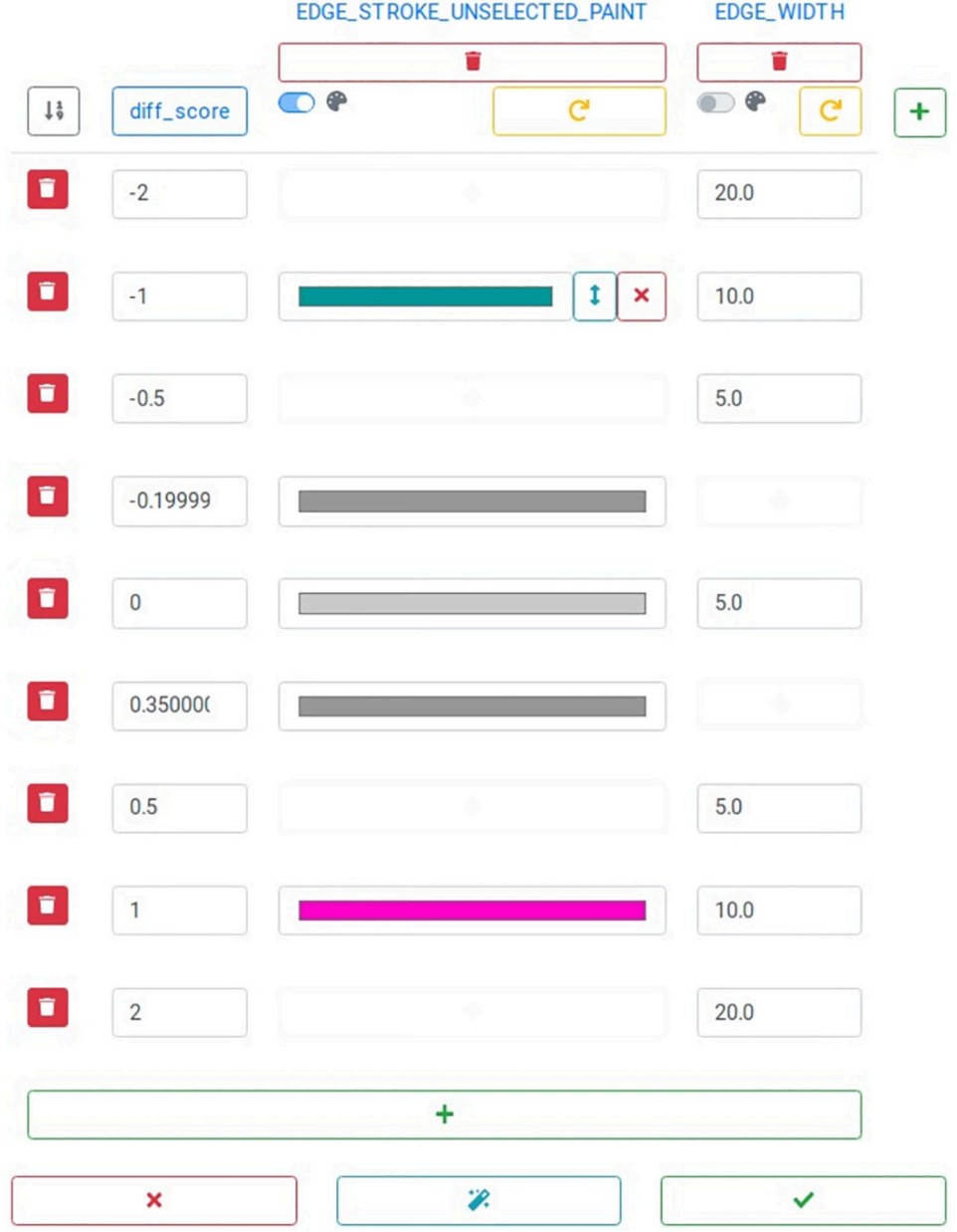

**Fig 3. Edit continuous mappings.** The score values of the edges are mapped using boundary values, to which colors and numeric values can be assigned. Mapping properties and boundaries can be deleted or new ones added.

the visual properties may not specify all for every visual property. New mapped values can be added using a gray plus button, which appears on moving the cursor over a blank field. Existing ones can not only be deleted but also moved within the visual property by the double-sided arrow next to it.

New thresholds can be added with the green plus button at the bottom. This will lead to the new value being attached at the end of the list, therefore the thresholds can be sorted by value. The single thresholds, and corresponding mapping values, can be deleted by the trash bin button next to it. The addition and removal of the visual properties work as for discrete mappings.

To facilitate the definition of continuous mappings for an attribute, a histogram of the contained data is displayed along with the editing form, as shown in Fig 4A) for the "diff_score" attribute. It can be seen, that the values lie in the range of -1 and +1. The bin size can be adjusted to get a better overview of the data. This histogram is also shown when the creation of the mapping is finished. Additionally, the different visual properties can be selected to display the resulting mapping. For mappings to colors this shows the corresponding color gradient with marked thresholds (Fig 4B), while for numerical values a graph of the mapping function is displayed (Fig 4C).

**Pass-through mapping.**   Pass-through mappings, as the name suggests, only pass the values of a property through to the mapping attribute. A relatable example is the labels of nodes that are displayed along. Although this mapping could be used to set other mapping properties, such as the node size, this way, in most cases it would be more appropriate to create a continuous mapping, which grants more flexibility afterward.

**Default properties.**   Mappings can only be created based on the data, which limits the visual representation of the network to the available data. Furthermore, general visual features need to be defined, like the background color of the network. For nodes, edges, and networks those properties can be set there, and then are consequently used as default values to decorate the networks. They also serve as a fallback when nodes and edges are not covered by the data used for the mappings.

## Graph layout

Cytoscape saves the coordinates of the nodes within the network in a dedicated aspect. However, this aspect is only optional, and even not all nodes must have coordinates provided. NDExEdit provides a variety of layout algorithms (Fig 5) to apply to a network, each with a special focus on the networks:

- **random**: nodes are distributed randomly across the viewport which enables to roughly explore the network and its content

- **grid**: nodes are arranged in a grid sorted by the node ids, which puts focus on the nodes

- **circular**: nodes are arranged in a circle so that the focus lies on the edges between the nodes

- **concentric**: nodes are arranged in concentric circles which is a more dense representation than the circular layout

- **hierarchical**: breadth-first arrangement of the network illustrates the topology of the network

- **force-driven**: cose (Compound Spring Embedder) layout [15] uses a physics simulation to determine node distances and produces a more dense representation of the network topology

- **preset**: initial layout saved within the network allows its restoration

## A) histogram of attribute values

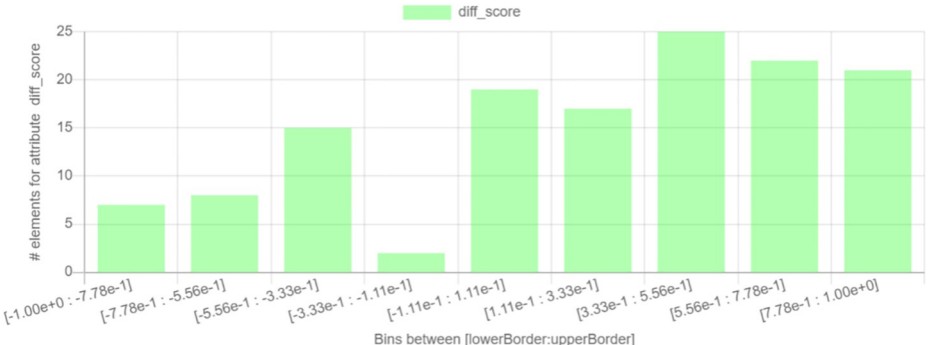

## B) color gradient for edge color mapping

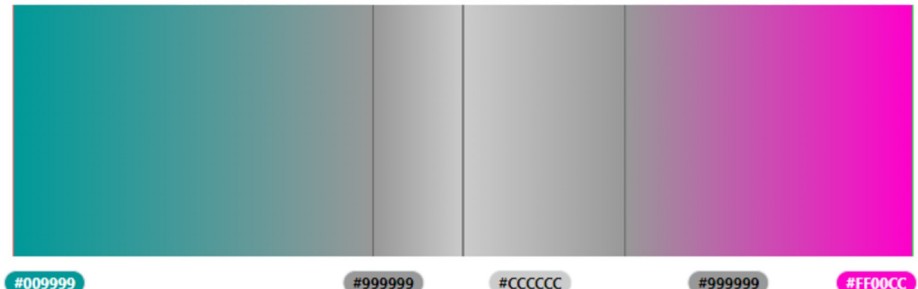

## C) edge width mapping graph

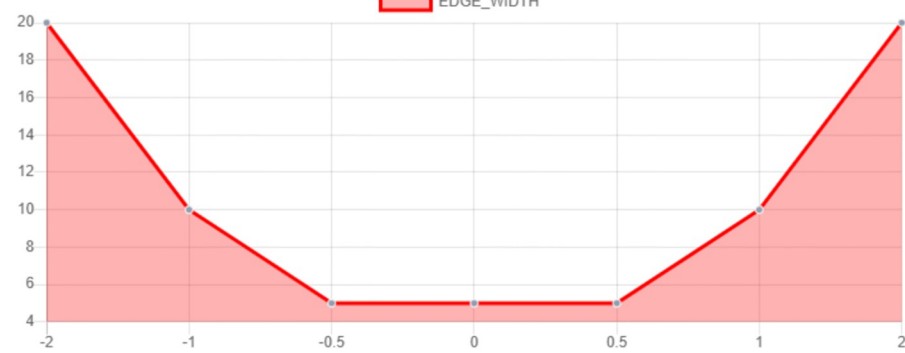

**Fig 4. Continuous mapping of values of the edge attribute "diff_score".** A) Histogram for the "diff_score" attribute values. B) Continuous mapping of the values to a color gradient with marked boundary values. C) Mapping graph for "diff_score" values to edge width.

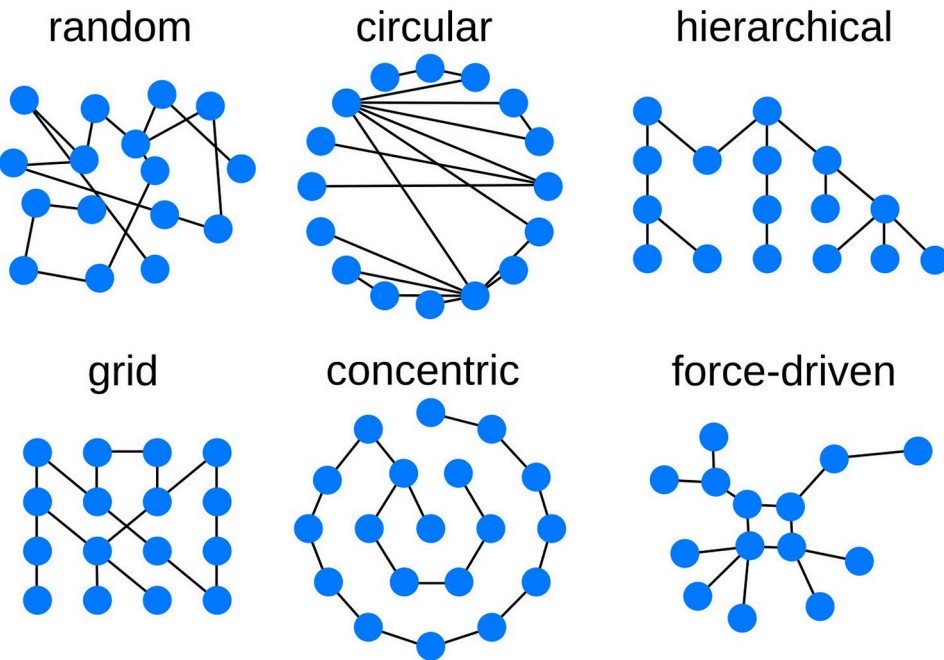

**Fig 5. Available graph layout options.** Different layout algorithms determine the position of each node, and therefore the overall representation of the network.

The final network layout, including manual refinements, is saved in the cartesian-layout aspect of the exported CX file and is therefore available for the subsequent usage of the network.

## Export

NDExEdit provides an option to export the modified networks, including their layout, and visual properties, and mappings as a compatible CX file. With provided credentials the networks can be directly exported to the NDEx platform, either creating a new network or updating an existing one. Also, the network can be exported as a compatible CX file that can be used by other applications.

Additionally, images in standard formats like PNG and JPEG can be created including a scaling factor to produce more detailed versions than a simple screen capture would allow. Also, the exported image can be set to only capture the viewport, or limited in its dimensions. For images in PNG format, it is also possible to change the background color or leave it transparent.

## Differentiation to Cytoscape

Cytoscape not only is a software tool for the visualization of networks, but moreover, it is a platform for data integration and analysis, supported by many third-party plugins. The focus of NDExEdit lies instead on the quick and simple visualization of networks based on the contained data. After an analysis workflow, the networks typically contain all the integrated information, and NDExEdit enables to explore its distribution and apply data-dependent mappings to create different visualizations.

Before mentioned workflows are often performed by processing, analyzing, and integrating the data in different tools, or programming languages like R or Python. Especially in the latter

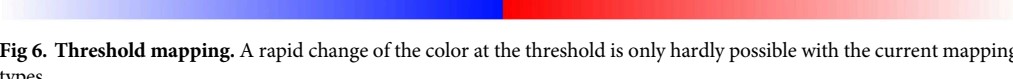

**Fig 6. Threshold mapping.** A rapid change of the color at the threshold is only hardly possible with the current mapping types.

visualizing the networks is tedious to perform programmatically. NDExEdit, therefore, offers a lightweight interface to generate visualizations. Furthermore, with the NDEx platform as a repository for the networks, collaborators can contribute and refine the final layout simply in the web browser.

Like many software, Cytoscape needs to be installed on local machines, which either requires the administrative rights of the user or has to be managed by the administrator of the institution, along with its software dependencies. This always causes security risks and vulnerabilities, if not handled carefully. An alternative provides web-based solutions, which also can be managed in a centralized manner. NDExEdit runs only in the web browser, without the need for any backend for data processing. This simplifies maintenance of private installations, which are indispensable within systems with limited internet access.

## Future directions

While inspecting several public networks missing features for mappings in general appeared: currently there is no elegant way of defining a mapping, that changes the color at a threshold (Fig 6). Currently, networks resemble this feature by defining a continuous mapping with two close, or even identical values as thresholds. The latter implicates further issues in the validation of the mapping.

On NDExEdit the specified mappings apply to the whole network, while it would be useful to restrict the mapping to certain sub-networks. Consequently, different mappings could be defined in general and switched on demand by the user. In the CX-format, as well as Cytoscape there already exists a possibility to manage different mappings for sub-networks and views. However, adaption on NDExEdit would require drastic adjustments to the used library for mapping the CX-format to Cytoscape.js.

Taking the idea of managing different mappings even further, would be the possibility to import existing mappings from other networks. This is possible in general, simply by manually editing the CX file and switching the "cyVisualProperties" aspect, but to be able to do it within NDExEdit would further improve the application. This also can be extended to an option to apply predefined visualization templates, such as SBGN [16], STRING [17], or Reactome [18, 19] layouts to a network.

While NDExEdit is intended to be a web application to easily change the visualization of the network dependent on the data, occasionally it would be beneficial to create additional data. For example, if the node degree is not provided as a property, it must be created with other tools to be available for mappings. More general, importing additional attributes from tabular data, or even the option to create whole networks from it can further decrease the barrier to create data-dependent visualizations of network data.

## Author Contributions

**Conceptualization:** Florian Auer, Frank Kramer.

**Funding acquisition:** Frank Kramer.

**Methodology:** Florian Auer, Simone Mayer.

**Project administration:** Frank Kramer.

**Software:** Florian Auer, Simone Mayer.

**Supervision:** Frank Kramer.

**Validation:** Florian Auer.

**Visualization:** Florian Auer, Simone Mayer.

**Writing – original draft:** Florian Auer.

**Writing – review & editing:** Frank Kramer.

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
