## [Decision Letter · Decision Letter 0]

27 Dec 2021

Dear Mr. Auer,

Thank you very much for submitting your manuscript "Data-dependent visualization of biological networks in the web-browser with NDExEdit" for consideration at PLOS Computational Biology.

As with all papers reviewed by the journal, your manuscript was reviewed by members of the editorial board and by several independent reviewers. In light of the reviews (below this email), we would like to invite the resubmission of a significantly-revised version that takes into account the reviewers' comments.

We cannot make any decision about publication until we have seen the revised manuscript and your response to the reviewers' comments. Your revised manuscript is also likely to be sent to reviewers for further evaluation.

Sincerely,

Dina Schneidman

Software Editor

PLOS Computational Biology

Reviewer's Responses to Questions

**Comments to the Authors:**

Reviewer #1: The NDExEdit application addresses an unmet need in the bioinformatics community today for data-dependent network visualization capabilities on the web. The authors present a functional application that implements the core style editing capabilities and is interoperable with Cytoscape and its associated NDEx Network Data Exchange repository via the Cytoscape CX standard data format. NDExEdit takes a significantly different approach compared to Cytoscape, one in which the interface and workflow is organized around the data - “what do you want to do with this node attribute” - rather than “I want to set the node color”. It integrates tools to review the distribution of data values for a given attribute to consult while choosing visual styles. Regardless of whether a given user would prefer one approach to the other, NDExEdit is not simply a Cytoscape look-alike or even work-alike.

It will be very useful that NDExEdit supports image export with controls on resolution and background color. Also, the English - Deutsch language option is welcome, as is support for multiple continuous mapping control points.

Here are several primary concerns, followed by other minor concerns:

The authors do not make the case for the significance and novelty of NDExEdit. They are selling themselves short. While it was important to describe Cytoscape and NDEx in the introduction, they do not describe all of the advantages of the workflows that are made easier on the web or in some cases only possible on the web. Their conclusion is also very brief and again, does not detail the advantages. In fact, there are valuable use cases that they do not even mention. Further, they do not contrast and compare their interface and workflow design with Cytoscape and discuss why some users might prefer their approach.

As a practical matter, the application itself does not enable the full workflow of opening a network from NDEx, editing it, and saving the modified network as an update or as a new network. Opening a network from NDEx is only supported by cut-and-paste of network UUIDs. Saving a network to NDEx is only possible by exporting it as a CX file and then uploading the file to NDEx. To our mind, even for light usage, it would be easier to install Cytoscape and access NDEx from the integrated interface. An interface to browse NDEx networks for import would not need to be complex - for example, the Cytoscape-NDEx interface is very straightforward. The authors acknowledge the missing “save” capability in their “Limitations and further improvements” section, stating that this is not possible because their application is not hosted at the same domain as NDEx. In fact, the two applications do not need to be at the same domain. For example, the NAGA web application at http://nbgwas.ucsd.edu/0.2.1/ (https://github.com/idekerlab/NBGWAS-Frontend) provides NDEx sign-in and save capability while being hosted at the UCSD domain.

An additional workflow outside the stated aims of the application should also be discussed: loading of tabular interaction data. A critical question is “where did the network that you load from NDEx come from?” NDEx does not currently have an interface for tabular upload and so networks in NDEx derived from data, such as the demo network in NDExEdit, are typically created in Cytoscape. But if the user already has the network in Cytoscape, why wouldn’t they just apply visual styling and layout there? One answer might be that a user might process their data in R or Python, generate their network, and load it directly into NDEx. The authors should discuss the full workflow - data to final network - that they have in mind.

Other minor concerns:

The CX generated by downloading the demo network fails validation when uploaded to NDEx. The original network published by Kim et al. can be exported from NDEx and re-uploaded successfully.

Error parsing element in CX stream: Expecting new aspect fragment at line: 1, column: 672066

When style mappings or other edits are disabled (shown with the lock icon) there is no explanation of *why* they are locked. Or in some cases, the lock icon is not visible because it has scrolled offscreen. This is very confusing.

Refreshing the browser window leads to a 404 error

Discrete mapping on edge color failed :

Reviewer #2: The paper presents a web-based library for drawing biological networks. The library is build upon cytoscape.js, it extends it with features for interactive setting of visual mapping functions for edges and nodes.

The paper presents the function of the library. The paper is well written and easily readable.

The paper describes the features of the library, however leaves unclear how this library outperforms other existing libraries for drawing networks online, e.g. Gosling.js, G6.sj, webcola.js, or other libraries.

The visual mapping allows the user to interactively set the mapping values. For colors, especilaly, it would be interesting which perceptually-linear interpolation scheme was used and how it is ensured that the color scheme is consistent with perception of colors.

Although abstract mentions multi-user environment, the core of the paper does not detail on this important aspect.

The source code is available on github.

The paper does not mention how the library deals with scalability - large values. What is the performance of the library in contrast to other existing libraries? such evaluation as well as user study is required for publication.

**Have the authors made all data and (if applicable) computational code underlying the findings in their manuscript fully available?**

Reviewer #1: Yes

Reviewer #2: Yes

PLOS authors have the option to publish the peer review history of their article (what does this mean?). If published, this will include your full peer review and any attached files.

Reviewer #1: **Yes: **Dexter Pratt and Trey Ideker

Reviewer #2: No
---

## [Decision Letter · Decision Letter 1]

15 May 2022

Dear Mr. Auer,

We are pleased to inform you that your manuscript 'Data-dependent visualization of biological networks in the web-browser with NDExEdit' has been provisionally accepted for publication in PLOS Computational Biology.

Best regards,

Dina Schneidman

Software Editor

PLOS Computational Biology

Reviewer's Responses to Questions

**Comments to the Authors:**

Reviewer #3: That was a dramatic improvement in the application. Thank you for being so responsive!

**Have the authors made all data and (if applicable) computational code underlying the findings in their manuscript fully available?**

Reviewer #3: Yes

PLOS authors have the option to publish the peer review history of their article (what does this mean?). If published, this will include your full peer review and any attached files.

Reviewer #3: No

---

## [Editor Report · Acceptance letter]

1 Jun 2022

PCOMPBIOL-D-21-02015R1 

Data-dependent visualization of biological networks in the web-browser with NDExEdit

Dear Dr Auer,

I am pleased to inform you that your manuscript has been formally accepted for publication in PLOS Computational Biology. Your manuscript is now with our production department and you will be notified of the publication date in due course.

With kind regards,

Zsofia Freund
